# SCENE: Self-Labeled Counterfactuals for Extrapolating to Negative Examples

**Deqing Fu    Ameya Godbole    Robin Jia**
University of Southern California
{deqingfu, ameyagod, robinjia}@usc.edu

## Abstract

Detecting negatives (such as non-entailment relationships, unanswerable questions, and false claims) is an important and challenging aspect of many natural language understanding tasks. Though manually collecting challenging negative examples can help models detect them, it is both costly and domain-specific. In this work, we propose **S**elf-labeled **C**ounterfactuals for **E**xtrapolating to **N**egative **E**xamples (SCENE), an automatic method for synthesizing training data that greatly improves models' ability to detect challenging negative examples. In contrast with standard data augmentation, which synthesizes new examples for existing labels, SCENE can synthesize negative examples zero-shot from only positive ones. Given a positive example, SCENE perturbs it with a mask infilling model, then determines whether the resulting example is negative based on a self-training heuristic. With access to only answerable training examples, SCENE can close 69.6% of the performance gap on SQuAD 2.0, a dataset where half of the evaluation examples are unanswerable, compared to a model trained on SQuAD 2.0. Our method also extends to boolean question answering and recognizing textual entailment, and improves generalization from SQuAD to ACE-whQA, an out-of-domain extractive QA benchmark.

## 1 Introduction

Many natural language understanding tasks require a model to distinguish claims that are supported by available evidence (i.e., *positive* instances) from ones that are not (i.e., *negative* instances). In question answering, unanswerable questions (*negative*) can be subtly different from answerable ones (*positive*)—for instance, inserting an unmentioned entity into an answerable question can make it unanswerable. In recognizing textual entailment (RTE) or fact verification, a hypothesis or claim that is not entailed by a given premise (*negative*) can be very

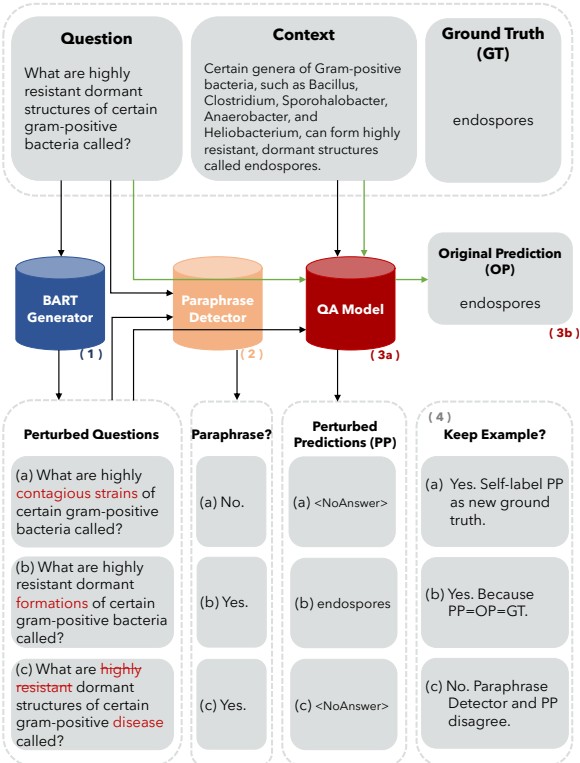

Figure 1: The *Self-labeled Counterfactuals for Extrapolating to Negative Examples (SCENE)* pipeline: (1) question perturbation using a mask in-filling model (BART); (2) paraphrase detection on perturbed questions; (3) QA model prediction for perturbed questions (3a) and the original question (3b); and (4) answer relabelling or filtering based on (2) and (3) (details in §3.2). Accepted new examples (generated via black arrows) are used for training in the same way as the original examples (green arrows).

similar to one that is entailed (*positive*). Training models to understand these fine distinctions remains an important open problem (Kim et al., 2021; Asai and Choi, 2021).

Collecting human-written negative examples, as done by datasets like SNLI (Bowman et al., 2015), MNLI (Williams et al., 2018) and SQuAD 2.0 (Rajpurkar et al., 2018), can yield training data that

helps models detect negatives. However, not only is this process expensive and time-consuming, but asking humans to write negative examples can also introduce dataset biases (Gururangan et al., 2018). Distant supervision (e.g., pairing questions with paragraphs that do not contain the answer to the question) can create negative examples at no additional annotation cost (Karpukhin et al., 2020; Lee et al., 2021), but the resulting examples are often simple for models. Training only on such examples does not prepare models for subtler negative examples that only differ in a small way from positive examples (e.g., see Table 1). Example (a) in Figure 1 provides a subtle unanswerable question by altering the phrase "resistant dormant structure" to "contagious strains." Such an edit keeps the question very related to the context but changes its meaning so that within the same context, the new question is no longer answerable.

We propose a new approach named *SCENE* (**S**elf-labeled **C**ounterfactuals for **E**xtrapolating to **N**egative **E**xamples) to automatically generate subtly negative examples given a dataset of only positive examples. We use these synthetic examples as training data to help a model recognize real negative examples zero-shot, thus avoiding the cost of collecting negative examples manually. Our approach first perturbs existing positive examples randomly using a mask-denoising model (BART), then uses self-training to dynamically label these perturbed examples as *negative* based on the model's current predictions (see Figure 1). To get self-training started, we also include negative examples generated by distant supervision to "warm-start" the model (i.e. train the model to predict the *negative* class). Unlike previous work (Min et al., 2020; Ross et al., 2022; Wu et al., 2021; Howard et al., 2022) that uses perturbations or counterfactuals for data augmentation, we use augmentation for *extrapolation*: we generate an entirely new class of examples not seen in the original training set.

Synthetic negative examples generated by SCENE teach models to recognize *real* negative examples. For extractive question answering (QA), we can extrapolate from SQuAD 1.1 (Rajpurkar et al., 2016), a positive-only dataset containing no unanswerable questions, to SQuAD 2.0 (Rajpurkar et al., 2018), which contains human-written unanswerable questions: our approach closes 69.6% of the gap with respect to a model trained on the SQuAD 2.0 training set. Our method using SQuAD

1.1 even outperforms a SQuAD 2.0-trained model when evaluated on the out-of-domain test set ACE-whQA (Sulem et al., 2021). SCENE can also extrapolate from BoolQ (Clark et al., 2019), a dataset containing only Yes/No questions, to BoolQ-3L (Sulem et al., 2022) which also contains "unanswerable/IDK" examples, closing 89.3% of the gap with a model trained on BoolQ-3L. It further applies to RTE (Wang et al., 2018), where we start with only entailment examples and synthesize non-entailment examples, closing 56.1% of the gap with a model trained on the full training set. Overall, our results show that automatically generated perturbations can be useful not only for increasing the amount of training data, but for enabling extrapolation to previously unseen types of examples.

## 2  Problem Setup

**Extractive QA.**  We first describe our setup for extractive QA, then modify it for other tasks. Our training data $\mathcal{D}_{\text{Positive}}$ only contains *positive* examples where each question is answerable, i.e. examples $(q, p, y)$ where $y$ is the answer to question $q$ on passage $p$. In particular to extractive QA, $y$ is a span in $p$. To extrapolate to the new unanswerable class, we aim to synthesize unanswerable examples zero-shot beyond existing baselines that generate simple unanswerable ones (see §3.1). Then, we use our proposed perturbation-based self-training procedure, SCENE, to generate more challenging unanswerable questions (see Figure 1 and §3.2).

We use RoBERTa$_{\text{Base}}$ (Liu et al., 2019) as our backbone QA model with parameters $\Theta$, denoted as $f_{\Theta}$ where for any possible answer $y'$, $f_{\Theta}(y', q, p) = \mathbb{P}(y' \mid p, q; \Theta)$. Sometimes we drop the parameter $\Theta$ or shorten to $f(q, p)$ to denote the probability output from QA model.

We adopt the standard procedure in identifying unanswerable questions in extractive QA tasks (Devlin et al., 2019): the model prediction $f(q, p)$ gives the probability for the answer's start and end positions over the entire span of the sequence, and the 0-position is always the `[CLS]` token. If the model has the highest probability in predicting `[CLS]`, we treat it as predicting unanswerable.

**Other tasks.**  For tasks beyond extractive QA, we modify notation and representations slightly. For boolean QA, the unanswerable (IDK) examples are represented as an additional class, on top of the original binary (Yes/No) classes. For

RTE, we represent the hypothesis-premise pair as $(h, p) \in \mathcal{D}_{\text{Positive}}$ with label $y = $ entailment. The generated negative examples belong to the new class with label $y = $ not entailment. We also use RoBERTa$_{\text{Base}}$ as our baseline model.

# 3 Method

We first present our method for extractive QA, then show how it applies to a broader set of tasks in §3.4. Our synthetic data generation process consists of two steps: the first is dedicated to generate easy unanswerable question-passage pairs; and the second is to generate harder unanswerable examples[1] through question perturbation and self-labelling.

## 3.1 Baselines for Generating Negatives

In §3.2, we will use our model to self-label perturbed questions as unanswerable. For this to work, we first need to introduce the concept of unanswerable questions to the model. Thus, we present two methods to create simple unanswerable examples.

**Shuffle-Based Generation.** The easiest way of synthesizing unanswerable question-passage pairs is through random pairing. To do this efficiently, we randomly pair passages and questions within a batch. Given a batch of question-passage pairs $\{(q_1, p_1), \cdots, (q_m, p_m)\}$, we randomly sample an element $\sigma$ from the permutation group $\mathbb{S}_m$, and reshuffle the pairs to be $\{(q_{\sigma(1)}, p_1), (q_{\sigma(2)}, p_2), \cdots, (q_{\sigma(m)}, p_m)\}$. For every $k$, we label the pair $(q_{\sigma(k)}, p_k)$ to be unanswerable if $\sigma(k) \neq k$; otherwise, we discard the example (since it is already present in the original batch). An example is shown in Table 8. $\ell_{\text{Shuf}}$ denotes the cross-entropy loss on shuffle-generated negatives.

**Retrieval-Based Generation.** To create harder unanswerable examples, given a question $q$, we retrieve a passage with high word overlap from the pool $\mathcal{P}$ of all passages in the dataset i.e. $\mathcal{P} = \{p | (q, p, y) \in \mathcal{D}_{\text{train}}\}$. Given an answerable example $(q, p, y)$, we create an unanswerable example $(q, R(q), \text{NoAns})$ where $R$ is the retrieval operator. In particular, $R(q)$ returns the passage from $\mathcal{P}$ that does not contain the answer string $y$ with the highest BM25 similarity (Robertson and Zaragoza, 2009) to the question $q$. $\ell_{\text{Retr}}$ denotes the cross-entropy loss on retrieval-generated negatives.

---

[1] We consider "harder/subtle unanswerable examples" as unanswerable questions that have high lexical overlap with original answerable questions, even though they may not be linguistically harder or syntactically diverse.

## 3.2 Self-Labeled Counterfactuals (SCENE)

The previous methods generate negative examples that are too easy, and thus do not teach the model to recognize more subtle negative examples (see evaluation in §4.2). To generate harder negative examples, we introduce **S**elf-labeled **C**ounterfactuals for **E**xtrapolating to **N**egative **E**xamples (SCENE). Given an example $(q, p, y)$ from our positive-only dataset $\mathcal{D}_{\text{Positive}}$, we use a generator $G$ to synthesize a perturbed version $G(q)$ of the question $q$. We then impute a label $\check{y}$, which is often the "unanswerable" label, and train the QA model to output $\check{y}$ given the input $(G(q), p)$. Prior work (Bartolo et al., 2021; Ross et al., 2022) synthesizes questions based on passages but, unlike our approach, cannot introduce a distribution shift to generate examples from a novel unanswerable class.

Inspired by self-training and counterfactual data augmentation (Howard et al., 2022), our method synthesize questions in three steps: Perturb, Filter, and Self-label. Figure 1 illustrates our method, and selected examples are shown in Tables 1 and 9.

**Perturb.** We first randomly mask some tokens from the question $q$ and use a mask-denoising model to fill the masks. Specifically, the proportion $\alpha$ of words to be masked is randomly drawn from a $\text{Beta}(2, 5)$ Distribution, and we use BART$_{\text{Large}}$ (Lewis et al., 2020) as the mask-denoising model.

**Filter.** Before we self-label the perturbed questions and update our model on them, we first filter out perturbations for which we have a lot of uncertainty about the correct label. Let $\delta_{\text{Pert}}(p, q, y) \in \{0, 1\}$ be the indicator variable for whether we use the perturbed pair $(G(q), p)$ or filter it out. Given the original example $(q, p, y)$ and perturbed question $G(q)$, we compute the model prediction $\hat{y} = \arg\max_{y'} f(y', q, p)$, and the perturbed prediction $\tilde{y} = \arg\max_{y'} f(y', G(q), p)$. In order to better determine whether the prediction $\tilde{y}$ is likely to be correct, we adopt the idea of rejection sampling by using a paraphrase detection model $\Gamma$—a RoBERTa$_{\text{Base}}$ model pre-trained on QQP from GLUE (Wang et al., 2018)—to help determine whether to filter out this example.

For a synthetic example $(G(q), p)$, we *discard* it (i.e., set $\delta_{\text{Pert}}(p, q, y) = 0$) if one of the following two cases happen: (1) *ambiguity*: $\Gamma(G(q), q) = $ Paraphrase but the perturbation changes the model prediction, i.e. $\hat{y} \neq \tilde{y}$. This suggests that we have contradictory conclusions

| | Question | Context | Answer |
|---|---|---|---|
| *Original* | What state is American Idol contestant Chris Daughtry from? | Ten of the fourteen Idol winners have come from the Southern United States [......] including Clay Aiken, Kellie Pickler, and Chris Daughtry, who are all from North Carolina. | North Carolina |
| *Perturbed* | What state is American Idol contestant Audrina Patridge and Chris Daughtry from? | | NoAns |
| *Original* | What is the downfall of using immunization against the pathogens that cause disease? | Immunization against the pathogens that cause diarrheal disease is a viable prevention strategy, however it does require targeting certain pathogens for vaccination. | require targeting certain pathogens for vaccination |
| *Perturbed* | What is the long-term goal of using immunization against the pathogens that cause disease? | | prevention strategy |

Table 1: Synthetic examples generated through perturbation and relabelling. The first example shows a perturbation induced unanswerable by inserting an unmentioned entity, and the second shows an induced answerable example but with a different answer due to a meaning-altering perturbation.

from the paraphrase detector and the QA model, and we cannot self-label it with confidence. (2) *bad prediction*: if the perturbation doesn't change the QA model prediction but they're both wrong, i.e. $\hat{y} = \tilde{y} \neq y$. We don't self-label with wrong labels. Note that if the QA model prediction stays the same (i.e. $\hat{y} = \tilde{y}$) while the paraphrase detector predicts $\Gamma(G(q), q) = \texttt{NotParaphrase}$, the perturbed question still passes the filter because non-paraphrase questions can still have the same answer.

**Self-Label.** If a synthetic example passes the filter, i.e., $\delta_{\text{Pert}}(p, q, y) = 1$, we trust the QA model's decision on the synthetic labels by self-labeling example $(G(q), p)$ with label $\tilde{y}$.

The batched objective for SCENE is denoted as follows, where $\ell(\cdot, \cdot)$ is the cross-entropy loss.

$$\ell_{\text{SCENE}}(\{(q_k, p_k, y_k)\}_{k=1}^{m})$$
$$= \frac{1}{m} \sum_{k=1}^{m} \delta_{\text{Pert}}(p_k, q_k, y_k) \cdot \ell(f(G(q_k), p_k), \tilde{y}_k) \quad (1)$$

### 3.3 Training

Putting everything together, we train on the weighted sum of the normal training objective $\ell$ and synthetic unanswerable objectives defined in previous sections. We denote the batched version of the weighted training objective as follows:

$$\ell_{\text{Overall}}(\{(q_k, p_k, y_k)\}_{k=1}^{m})$$
$$= \frac{1}{m} \sum_{k=1}^{m} \ell(f(q_k, p_k), y_k) + \lambda_{\text{SCENE}} \cdot \ell_{\text{SCENE}}(\{(q_k, p_k, y_k)\})$$
$$+ \lambda_{\text{Shuf}} \cdot \ell_{\text{Shuf}}(\{(q_k, p_k, y_k)\}) + \lambda_{\text{Retr}} \cdot \ell_{\text{Retr}}(\{(q_k, p_k, y_k)\}) \quad (2)$$

where $\lambda_{\text{Shuf}}$, $\lambda_{\text{Retr}}$ and $\lambda_{\text{SCENE}}$ denote weights for their corresponding losses.

Note that we perform SCENE data augmentation for every batch during training using the current

model parameters $\Theta$—we do not use a frozen pretrained model to statically perform SCENE augmentation just once at the beginning. To make sure the predictions $\hat{y}$ and $\tilde{y}$ are mostly correct with respect to their questions, the initial weights for our QA model $f_{\Theta}$ are pre-trained on the positive-only dataset $\mathcal{D}_{\text{Positive}}$. For warm-starting purposes (see §3.1), $\lambda_{\text{SCENE}} = 0$ for the first few steps (see details in §B.2). The rest of the training follows the objective in Eq 2. For the entire training procedure, our method is *not* provided with any human-annotated negatives, and therefore methods such as early stopping, model selection, etc. cannot be used.

### 3.4 Beyond Extractive Question Answering

Our method can also work beyond extractive QA tasks. SCENE can work on boolean QA in a setting where we extrapolate from BoolQ (Clark et al., 2019), a collection of question-passage pairs $(q, p)$ of label $y =$ Yes or No, to BoolQ-3L (Sulem et al., 2022), an extended dataset to BoolQ with additional unanswerable (IDK) examples. We repeat the SCENE pipeline (Perturb, Filter and Self-label), together with Shuffle, and evaluated on BoolQ-3L test set. Because the BoolQ-3L's IDK questions were produced using information retrieval similar to what we did in §3.1, we refrain from using retrieval-based IDK questions during training.

Our method can also work on binary RTE in a setting where we only have access to hypothesis-premise pairs $(h, p) \in \mathcal{D}_{\text{Positive}}$ of label $y = \texttt{entailment}$. For RTE, we modify the SCENE pipeline so that we perturb the premise $p$ to $G(p)$. Instead of the original self-labeling step, we label $G(p)$ to be $\tilde{y} = \texttt{not entailment}$ if the paraphrase detector predicts that $G(p)$ is not a paraphrase of

$p$. We note that unlike for QA, SCENE can generate negative examples even without Shuffle or Retrieval-based negatives. The necessity of Shuffle for binary NLI tasks is ablated in experiments (see Table 7). We do not experiment with retrieval for RTE because the dataset contains only a small pool of hypotheses to use for retrieval.

# 4 Experiments

## 4.1 Experimental Details

We define a metric, *Performance Gap*, that measures how much of the gap between training on positives only and the full dataset (with human-written negatives) can be closed with SCENE:

$$\text{Gap} = \frac{\mathcal{M}[SCENE(\mathcal{D}_{\text{Positive}}^{\text{train}})] - \mathcal{M}[\text{Baseline}(\mathcal{D}_{\text{Positive}}^{\text{train}})]}{\mathcal{M}[\text{Baseline}(\mathcal{D}_{\text{Full}}^{\text{train}})] - \mathcal{M}[\text{Baseline}(\mathcal{D}_{\text{Positive}}^{\text{train}})]}$$
(3)

where $\mathcal{M}$ is any task-specific metric on the full test set $\mathcal{D}_{\text{Full}}^{\text{test}}$. The arguments of $\mathcal{M}$ specify the method and the training set.

### 4.1.1 Extractive QA

We use **SQuAD 1.1** (Rajpurkar et al., 2016) as the *training* set for extractive QA. It is a collection of question-passage-answer triples derived from Wikipedia articles, where the correct answers of questions can be any sequence of tokens in the given text. It is split into 87,599 training examples and 10,570 validation examples.

**SQuAD 2.0** (Rajpurkar et al., 2018) is the *testing* set. It is an extended version of SQuAD 1.1 with additional unanswerable questions. We evaluate on the development set, which contains 11,873 examples—5,928 examples with answers and 5,945 unanswerable examples. When evaluating on SQuAD 2.0, the metrics of interest are Exact Match ($EM$) accuracy and $F_1$ scores.

**ACE-whQA** (Sulem et al., 2021) is the out-of-domain *(OOD) testing* set for extractive QA, derived from an event extraction corpus, ACE (Walker et al., 2006). Aside from being an OOD test set for SQuAD 2.0, ACE-whQA also distinguishes two types of unanswerable (IDK) questions: (1) *competitive*, where the passage includes an entity of the same type as the expected answer; and (2) *non-competitive*, where the passage does not include an entity of the same type. It contains 238 answerable examples, 250 competitive IDK examples, and 245 non-competitive IDK examples.

**A Probabilistic Alternative.** Instead of training with human-annotated negative examples, one can force the model to predict unanswerable by setting a probabilistic threshold, i.e., the model predicts unanswerable if $\mathbb{P}(\hat{y} \mid p, q; \Theta) < \theta_{\text{threshold}}$. However, one would need a dataset with negative examples first to find the best $\theta_{\text{threshold}}$. In our experiments, we compare SCENE with an oracle approach that uses the validation set of the target dataset to choose the best threshold.

### 4.1.2 Boolean QA

We use **BoolQ** (Clark et al., 2019) as the *training* set for boolean QA. It is a collection of question-passage-answer triples gathered from queries to the Google search engine. Unlike SQuAD, answers in BoolQ are either Yes or No. It is split into 9,427 training examples and 3,270 validation examples.

**BoolQ-3L** (Sulem et al., 2022) is the *testing* set. It is an extended version of BoolQ with additional IDK (unanswerable) examples generated through retrieval. The dataset contains 4,906 validation examples, where 33% of them are IDK examples. When evaluating on BoolQ-3L, the metric of interest is the classification accuracy across three categories (Yes/No/IDK).

### 4.1.3 Recognizing Textual Entailment

**RTE** (Wang et al., 2018) is a collection of hypothesis-premise pairs labeled as either "entailment" or "not-entailment." It has 2,490 training examples and 277 validation examples. For our method, we only use the 1,249 entailment subset from the training set as our *training* data $\mathcal{D}_{\text{Positive}}$. We still *test* on the full validation split.

## 4.2 Extractive QA Results

For comparison as baseline/oracle models, we train normally both on the positive-only training set $\mathcal{D}_{\text{Positive}}$ and on the full training set $\mathcal{D}_{\text{Full}}$. We evaluated on 3 different independent runs of training and report averages to mitigate the effects of randomness. For simplicity, we only let $\lambda$'s be zero or one, unless otherwise indicated.

**SCENE enables extrapolation to SQuAD 2.0.** Table 2 shows results on the SQuAD 2.0 evaluation set. The best SCENE method achieves 71.8 F1, which closes 69.6% of the gap in SQuAD 2.0 accuracy between the baseline that trains on SQuAD 1.1 (45.7 F1) and the oracle method that trains on SQuAD 2.0 (83.2 F1). Our best method combines retrieval with self-labeled counterfactuals; this is 7.5 F1 points better than only using retrieval. Using shuffle-based negative examples barely improves

| Training Set | Method | Has Answer | | No Answer | | Average | |
|---|---|---|---|---|---|---|---|
| | | $EM$ | $F_1$ | $EM$ | $F_1$ | $EM$ | $F_1$ |
| SQuAD 1.1 | No Augmentation | **84.4 (0.2)** | **91.5 (0.1)** | 0.0 (0.0) | 0.0 (0.0) | 42.1 (0.1) | 45.7 (0.1) |
| | Best Threshold | 39.8 (7.4) | 40.2 (7.7) | **68.8 (7.5)** | **68.8 (7.5)** | 54.4 (0.1) | 54.5 (0.1) |
| | Shuffle | 84.1 (0.3) | 91.3 (0.1) | 3.4 (0.3) | 3.4 (0.3) | 43.7 (0.1) | 47.3 (0.1) |
| | Shuf + *SCENE* | 80.1 (0.2) | 86.5 (0.1) | 40.8 (1.7) | 40.8 (1.7) | 60.4 (0.8) | 63.7 (0.8) |
| | Retrieval | 79.3 (0.1) | 85.5 (0.2) | 43.1 (0.7) | 43.1 (0.7) | 61.2 (0.4) | 64.2 (0.4) |
| | Retr + *SCENE* | 75.8 (0.6) | 81.4 (0.6) | 62.2 (0.9) | 62.2 (0.9) | **69.0 (0.2)** | **71.8 (0.1)** |
| SQuAD 2.0 | No Augmentation | 78.2 (0.2) | 84.6 (0.3) | 81.8 (0.5) | 81.8 (0.5) | 80.0 (0.2) | 83.2 (0.1) |
| | *SCENE* | 76.3 (0.3) | 82.2 (0.2) | 84.6 (0.4) | 84.6 (0.4) | 80.4 (0.1) | 83.4 (0.1) |

Table 2: **Evaluation on SQuAD 2.0**. Training on positive-only SQuAD 1.1, SCENE with Retrieval can close 69.6% of the gap compared with an oracle model trained on SQuAD 2.0. We report the mean and standard deviation (in parentheses) over 3 runs. Best scores among all methods that train on SQuAD 1.1 are highlighted in **bold**.

| Training Set | Method | Has Answer | | No Answer | | Average | |
|---|---|---|---|---|---|---|---|
| | | $EM$ | $F_1$ | Competitive $EM/F_1$ | Non-Comp $EM/F_1$ | $EM$ | $F_1$ |
| SQuAD 1.1 | No Augmentation | **79.1 (0.5)** | **86.8 (0.8)** | 1.1 (1.2) | 7.9 (7.8) | 41.8 (2.1) | 45.6 (1.9) |
| | Best Threshold | 56.2 (3.5) | 57.5 (3.7) | 68.9 (3.8) | 93.4 (2.1) | 68.7 (0.5) | 69.3 (0.5) |
| | Shuffle | 69.1 (1.8) | 78.0 (1.0) | 46.3 (3.5) | 76.3 (3.7) | 65.2 (1.8) | 69.7 (1.5) |
| | Shuf + *SCENE* | 59.0 (3.7) | 65.1 (4.9) | 69.7 (5.0) | 87.3 (2.7) | 68.7 (0.2) | 71.8 (0.6) |
| | Retrieval | 59.0 (0.2) | 65.4 (0.8) | 68.1 (1.8) | 86.9 (1.5) | 68.2 (0.9) | 71.4 (0.6) |
| | Retr + *SCENE* | 55.0 (4.2) | 61.7 (4.2) | **73.8 (4.4)** | **95.3 (0.4)** | **69.8 (0.9)** | **73.1 (0.9)** |
| SQuAD 2.0 | No Augmentation | 70.9 (0.8) | 78.6 (1.1) | 31.7 (4.5) | 60.2 (10.4) | 58.4 (3.3) | 62.3 (3.2) |
| | *SCENE* | 49.2 (2.3) | 56.9 (1.8) | 75.4 (2.1) | 91.3 (1.1) | 66.3 (0.7) | 70.1 (0.4) |

Table 3: **OOD Evaluation on ACE-whQA**. Trained only on SQuAD 1.1, SCENE with Retrieval even outperforms the oracle model trained on SQuAD 2.0. Applying SCENE also improves SQuAD 2.0-trained models out-of-domain. Best scores among all methods that train on SQuAD 1.1 are highlighted in **bold**.

over the SQuAD 1.1 baseline; adding self-labeled counterfactuals to this improves F1 by 16.4. Compared with setting probabilistic thresholds to force unanswerable predictions, our best method does not require additional annotated SQuAD 2.0 examples to find thresholds and still achieves higher performance. Finally, adding self-labeled counterfactuals to the SQuAD 2.0 training set does not significantly improve SQuAD 2.0 accuracy since synthetic unanswerable examples are not as strong as human-annotated in-distribution ones. Note that training on either gold-standard SQuAD 2.0 negatives or SCENE negatives reduces accuracy on the "HasAns" subset, as introducing the concept of unanswerability inevitably causes models to sometimes predict "NoAns" on answerable questions.

**SCENE improves out-of-domain generalization.** Table 3 shows results where we train on SQuAD-derived data and test on ACE-whQA. We focus on the overall average accuracy, where we average the accuracy on the "HasAns" and "NoAns" subsets of ACE-whQA. SCENE combined with retrieval and the positive-only SQuAD 1.1 dataset achieves the highest accuracy, even outperforming the oracle

model trained on SQuAD 2.0 (73.1 F1 vs. 62.3 F1). Adding SCENE examples to the SQuAD 2.0 training data improves the ACE-whQA accuracy by 7.8 F1, suggesting that including SCENE improves the overall utility of SQuAD 2.0 as training data.

**Qualitative results and statistics.** SCENE can synthesize subtly unanswerable questions in many ways, such as inserting or replacing unmentioned entities, lexical changes, and tense modifications. Tables 1 and 9 show some selected synthetic unanswerable examples generated via SCENE, and Table 10 shows randomly-sampled ones. In Table 11, we adopt different categories of negative examples defined by Rajpurkar et al. (2018) and compare their frequency in SCENE with SQuAD 2.0. SCENE tends to create more questions that add an impossible condition, and fewer antonym substitutions or negations since the BART model is not tuned particularly to produce these edits.

**Ablation study.** We conduct ablation study on our best method, which combines retrieval and self-labeled counterfactuals. We test several simplified versions of our filtering and relabeling pipeline:

| Method | | | SQuAD 2.0 | | ACE-whQA | |
|---|---|---|---|---|---|---|
| Shuffle | Retrieval | SCENE | $EM$ | $F_1$ | $EM$ | $F_1$ |
| No Augmentation | | | 42.1 (0.1) | 45.7 (0.1) | 41.8 (2.1) | 45.6 (1.9) |
| ✓ | | | 43.7 (0.1) | 47.3 (0.1) | 65.2 (1.8) | 69.7 (1.5) |
| ✓ | ✓ | | 61.3 (0.2) | 64.3 (0.2) | 69.4 (0.7) | 72.8 (0.5) |
| ✓ | | ✓ | 60.4 (0.8) | 63.7 (0.8) | 68.7 (0.2) | 71.8 (0.6) |
| | ✓ | | 61.2 (0.4) | 64.2 (0.4) | 68.2 (0.9) | 71.4 (0.6) |
| | ✓ | ✓ | 69.0 (0.2) | **71.8 (0.1)** | **69.8 (0.9)** | **73.1 (0.9)** |
| ✓ | ✓ | ✓ | **69.7 (0.4)** | 71.7 (0.2) | 63.4 (1.6) | 65.9 (1.6) |

Table 4: Ablation study on all combinations of negative example generation methods. All experiments are trained on SQuAD 1.1 and evaluated on SQuAD 2.0 and ACE-whQA. Best scores are highlighted in **bold**. Our best model uses the combination of Retrieval and SCENE.

**In-Domain Evaluation on SQuAD 2.0**

| Method | Has Answer | | No Answer | | Average | |
|---|---|---|---|---|---|---|
| | $EM$ | $F_1$ | $EM$ | $F_1$ | $EM$ | $F_1$ |
| Assume NoAns | 66.2 | 70.2 | **70.8** | **70.8** | 68.5 | 70.5 |
| ↑ w/o Retr | 71.3 | 76.0 | 58.8 | 58.8 | 65.0 | 67.4 |
| Only NoAns | **76.8** | **82.5** | 59.0 | 59.0 | 67.9 | 70.7 |
| No Filter | 69.9 | 74.9 | 65.1 | 65.1 | 67.5 | 69.9 |
| SCENE (ours) | 75.8 | 81.4 | 62.2 | 62.2 | **68.9** | **71.8** |

**Out-of-Domain Evaluation on ACE-whQA**

| Method | Has Answer | | No Answer | | Average | |
|---|---|---|---|---|---|---|
| | $EM$ | $F_1$ | Comp | Non Comp | $EM$ | $F_1$ |
| Assume NoAns | 23.9 | 27.5 | **96.0** | **99.2** | 60.8 | 62.6 |
| ↑ w/o Retr | 23.9 | 26.4 | 91.2 | 96.2 | 58.8 | 60.1 |
| Only NoAns | 43.0 | 49.4 | 85.3 | 98.2 | 67.4 | 70.6 |
| No Filter | 37.1 | 41.6 | 90.4 | 98.9 | 65.9 | 68.1 |
| SCENE (ours) | **55.0** | **61.7** | 73.8 | 95.3 | **69.8** | **73.1** |

Table 5: **Ablation study** on different filtering and labeling strategies for synthetic samples. The full SCENE pipeline achieves the best performance both in-domain and out-of-domain.

1. *Assume NoAns*: We accept every perturbed example and relabel everyone as unanswerable, i.e. $\forall (p, q, y), \delta_{\text{Pert}}(p, q, y) = 1$ and $\tilde{y} = $ NoAns.
2. *Assume NoAns w/o Retr*: Same as above, but without Retrieval to generate simple negatives.
3. *Only NoAns*: We additionally filter out perturbed questions for which our imputed label is *not* unanswerable, i.e., $\tilde{y} \neq$ NoAns. This tests whether the answerable questions generated by SCENE contribute to the performance.
4. *No Filter*: We do not filter and we perform self-training with synthetic examples $(G(q), p, \tilde{y})$.

Results of these ablations are shown in Table 5. Our full approach performs the best both in-domain and out-of-domain. Accepting all perturbations as unanswerable is surprisingly competitive on SQuAD 2.0, but is much worse on ACE-whQA,

as it encourages the model to predict NoAns too much. Counter-intuitively, both "Assume NoAns w/o Retr" and "Only NoAns" reduce accuracy on unanswerable questions in SQuAD 2.0. One possible explanation is that if all perturbations are unanswerable, detecting unanswerables becomes the task of detecting perturbations; also including perturbed answerable questions reduces this spurious correlation. Finally, removing the paraphrase detector decreases accuracy by 1.9 F1 on SQuAD 2.0 and 5.0 F1 on ACE-whQA, showing that our filtering strategy does help reduce noise, but self-training alone without filtering is still effective.

We also conduct ablation study on different combinations of losses, $\ell_{\text{Shuf}}$, $\ell_{\text{Retr}}$ and $\ell_{\text{SCENE}}$, presented in Table 4. Results suggest that our best model uses the combination of Retrieval and SCENE and performing Shuffle together with Retrieval can be redundant since negatives provided by Retrieval are a refined subset of examples provided by Shuffle.

### 4.3 Boolean QA Results

SCENE enables extrapolation to BoolQ-3L, as shown in Table 6. Note that to be consistent with our extractive QA experiments, we report the average of the accuracy on negative examples (IDK label) and positive examples (yes and no labels). The best SCENE method achieves 78.1% accuracy, which closes 89.3% of the gap in BoolQ-3L accuracy between the baseline that trains on BoolQ (38.9% accuracy) and the oracle method that trains on BoolQ-3L (82.8% accuracy). Our best method combines shuffle with self-labeled counterfactuals. Adding self-labelled counterfactuals to the BoolQ-3L training set slightly improves BoolQ-3L accuracy by 0.4 points.

| Train Set | Method | Positive | | Negative | Average |
| | | Yes | No | IDK | |
|---|---|---|---|---|---|
| BoolQ (Yes/No) | No Augment | **83.9** | 72.0 | 0.0 | 38.9 |
| | Shuffle | 79.9 | **73.6** | 78.4 | 77.5 |
| | Shuf + *SCENE* | 77.1 | 71.7 | **81.7** | **78.1** |
| BoolQ-3L (Yes/No/IDK) | No Augment | 80.9 | 73.5 | 88.4 | 82.8 |
| | *SCENE* | 82.9 | 70.5 | 89.7 | 83.2 |

Table 6: **Evaluation on BoolQ-3L**. SCENE with BoolQ alone achieves 89.3% of the performance of an oracle model trained on BoolQ-3L. Best scores among all methods that train on BoolQ are highlighted in **bold**.

| Training Set | Method | Accuracy |
|---|---|---|
| Entailment-Only Subset (RTE$_{EOS}$) | No Augmentation | 52.7 |
| | Shuffle | 54.2 |
| | *SCENE* | 64.3 |
| | Shuf + *SCENE* | **67.9** |
| Full Dataset | No Augmentation | 79.8 |

Table 7: **Evaluation on RTE.** SCENE with RTE$_{EOS}$ alone achieves 56.1% of the performance of an oracle model trained on RTE. Best scores among all methods that train on the RTE$_{EOS}$ are highlighted in **bold**.

## 4.4 RTE Results

SCENE also extends to binary RTE tasks by enabling extrapolation from the entailment-only subset to all of RTE, as shown in Table 7. The best SCENE method achieves 67.9% accuracy, which closes 56.1% of the gap in RTE accuracy between the baseline that trains on entailment-only subset (52.7% accuracy) and the oracle method that trains on full RTE (79.8%). Our best method combines shuffle and self-labeled counterfactuals. Using shuffle-based negative examples barely improves over the baseline that trains on the subset, and using self-labeled counterfactuals alone improves over the baseline by 11.6 points. Combining shuffle and self-labeled counterfactuals improves accuracy by 13.7 and 3.6 points respectively, compared to using each only their own.

## 5 Discussion and Related Work

**Self-Training.** Self-training is a semi-supervised learning approach that utilizes a teacher model to assign labels to unlabelled data, which is then used to train a student model (Yarowsky, 1995; Mc-Closky et al., 2006; Kumar et al., 2020). In our work, we adopt a self-training scheme where the teacher model and the student model are the same. During training, the current model (teacher) is used to annotate new examples for the next batch (student), and it is also used jointly with a paraphrase detector (as a rejection sampler) to reduce noise. SCENE also differs from standard self-training in that, rather than using a pool of unlabelled examples, we generate them.

**Hard Negative Mining.** Several NLP datasets have gathered instances of the negative class for their task. Some have relied on human annotation e.g., unsupported claims in fact verification (Aly et al., 2021; Wadden et al., 2020), non-entailed hypotheses in NLI (Bowman et al., 2015), unanswerable questions (Rajpurkar et al., 2018), *inter alia*. Some have used heuristics and external knowledge sources to automatically mine negative examples (Lee et al., 2021; Wright et al., 2022). Finally, there are hybrid approaches where candidate negative examples are first automatically gathered and then manually verified (Wadden et al., 2022). Our baseline approach for generating negatives based on in-batch negatives via shuffling and retrieval is similar in motivation to mining negatives for neural ranking models (Karpukhin et al., 2020).

**Counterfactual Data Augmentation.** Counterfactual data augmentation (Kaushik et al., 2020) has been studied as an approach to reduce model reliance on spurious correlations (Gardner et al., 2021) by creating minimal pairs that flip the label. Past work includes manual approaches such as Contrast Sets (Gardner et al., 2020) which asks expert annotators to minimally perturb examples, or heuristic approaches that rely on synonym-antonym sets to substitute words in the example (Wang and Culotta, 2021; Chen et al., 2021). Several approaches leveraging language models (LM) for generating counterfactuals have been proposed: Polyjuice (Wu et al., 2021) and Tailor (Ross et al., 2022) train LMs to generate minimal edit counterfactuals through specified control codes, LIT (Li et al., 2020) automatically generates contrasts sets for NLI tasks based on linguistic rules but lacks flexibility, and NeuroCounterfactuals (Howard et al., 2022) generates counterfactuals that are relevant but not restricted to be minimal edits by leveraging constrained decoding from task fine-tuned LMs. Despite their success in perturbing texts, it is unclear how to use these methods to introduce distribution shift and extrapolate to unseen label sets (in our case, negative classes). Our approach of using masked language models for text in-filling followed by filtering is inspired by recent

work on open set classification (Xu et al., 2022).

**Synthetic Question Generation.** Many approaches have been proposed to synthesize questions given a passage and an answer within it (Du et al., 2017; Du and Cardie, 2018; Lewis and Fan, 2019; Alberti et al., 2019; Bartolo et al., 2021; Lewis et al., 2021). These methods are designed to generate answerable question-answer pairs and do not directly apply to our setting where we must synthesize challenging unanswerable questions. Pan et al. (2021) generate (topically relevant) unanswerable questions by conditioning a question generator on phrases from the surrounding context of passages used in the QA dataset. SCENE does not assume access to such external text sources (e.g. source documents of QA datasets).

# 6 Conclusion

In this work, we have developed a counterfactual generation pipeline, SCENE, which synthesizes negative examples (as well as some positive examples) from real positive ones. The counterfactual generation is performed through text perturbation, filtering and relabelling. SCENE enables extrapolation to negative examples given only real positive examples, closing 69.6% of the performance gap on extractive QA, 89.3% on boolean QA, and 56.1% on RTE, compared with models trained on both real positive and negative ones.

In the future, we hope to combine our automatic pipeline with human annotation effort. For example, we could use adversarial data collection to collect a small number of perturbations that create challenging negative examples, and use these to guide generation of self-labeled counterfactuals. We would also like to explore ways to backpropagate signal from the filtering process into the generator, so the generator learns to generate useful perturbations. Overall, we hope that our work can inspire more work on how synthetic data can enable new types of model extrapolation.

## Limitations

Though our method alleviates data collection cost by human annotators, its computational cost is higher than training with human annotated datasets for multiple reasons (see Appendix B.3). Adding synthetic examples increases the time required for one epoch of training. Moreover, SCENE examples are generated with a BART model and filtered

by a paraphrase detector during training, both of which add computational overhead.

We only validated on extractive QA, boolean QA and RTE. Whether SCENE can be applied to other tasks that require detecting challenging negatives is unknown. SCENE is also limited to extrapolating to one pre-defined new class, *negative* examples; whether SCENE can be used to extrapolate to other types of classes is also unknown.

While Large Language Models (with billions of parameters) have demonstrated the ability to perform zero/few-shot generalization on unseen tasks, their performance still lags behind fine-tuned, task-specific models. Thus, our method and analysis focus on less expensive smaller language models as they are high-performing while being cheaper to evaluate and fine-tune. Moreover, the pre-training and fine-tuning data for Large Language Models is largely unknown. Additional human feedback is used for training models to abstain from answering certain queries (Ouyang et al., 2022). Since SCENE is an approach for training models to detect examples from an unseen negative class, we remove the confounding variable of unknown training data by focusing on language models with public training corpora without human intervention. Finally, it can be an unfair comparison with LLMs on open-web datasets like SQuAD due to data contamination concerns when LLMs are trained. Data contamination acknowledgements can be found at Section 8 from the PaLM technical report (Chowdhery et al., 2022) where SQuAD 2.0 is mentioned specifically as contaminated.. Nevertheless, we will consider LLMs for future research and how our proposed method would fit into the LLM cycles.

## Acknowledgements

We thank Ting-Yun Chang, Johnny Wei, Wang Zhu, and the members of USC NLP Group for their valuable feedback. This work was supported in part by an Open Philanthropy research grant and a Google Scholar Research Award.

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

# A Examples and Statistics

## A.1 Selected Examples

| | Original | Shuffle | Information Retrieval |
|---|---|---|---|
| **Question** | Zinc oxide is believed to be mentioned in what ancient text? | When did Seoul Semiconductor release the first high DC voltage LED? | Zinc oxide is believed to be mentioned in what ancient text? |
| **Context** | The Charaka Samhita, thought to have been written between 300 and 500 AD, mentions a metal which, when oxidized, produces pushpanjan, thought to be zinc oxide. Zinc mines at Zawar, near Udaipur in India, have been active since the Mauryan period. The smelting of metallic zinc here, however, appears to have begun around the 12th century AD. | The Charaka Samhita, thought to have been written between 300 and 500 AD, mentions a metal which, when oxidized, produces pushpanjan, thought to be zinc oxide. Zinc mines at Zawar, near Udaipur in India, have been active since the Mauryan period. The smelting of metallic zinc here, however, appears to have begun around the 12th century AD. | Roughly one quarter of all zinc output in the United States (2009), is consumed in the form of zinc compounds; a variety of which are used industrially. Zinc oxide is widely used as a white pigment in paints, and as a catalyst in the manufacture of rubber. |
| **Answer** | Charaka Samhita | No Answer | No Answer |

Table 8: Examples of *Simple* Unanswerable Questions using Shuffle or Retrieval.

| | Question | Context | Self-Labeled Answer |
|---|---|---|---|
| *Original* | In which video does it show Madonna being scolded by her boss in Italian? | Madonna's Italian-Catholic background and her relationship with her parents are reflected in the album Like a Prayer. [......] The "Open Your Heart" video sees her boss scolding her in the Italian language. On the Who's That Girl World Tour, she dedicated the song "Papa Don't Preach" to Pope John Paul II. | Open Your Heart |
| *Perturbed* | In what video does it sound like Madonna being scolded by her mother in Italian? | | NoAns |
| *Original* | What are highly resistant dormant structures of certain gram-positive bacteria called? | Certain genera of Gram-positive bacteria, such as Bacillus, Clostridium, Sporohalobacter, Anaerobacter, and Heliobacterium, can form highly resistant, dormant structures called endospores. Endospores have a central core of cytoplasm containing DNA and ribosomes surrounded by a cortex layer. | endospores |
| *Perturbed* | What are highly contagious strains of certain gram-positive bacteria called? | | NoAns |
| *Original* | When did the Everton club board fire Smith? | The Everton board finally ran out of patience with Smith and he was sacked in March 2002 after an FA Cup exit at Middlesbrough, with Everton in real danger of relegation. Everton qualified for the 2007–08 and 2008–09 UEFA Cup competitions and they were runners-up in the 2009 FA Cup Final. | March 2002 |
| *Perturbed* | When will the football club finally fire Smith? | | NoAns |
| *Original* | What did the Governor-General do with the first assent? | In Australia, a technical issue arose with the royal assent in both 1976. In 1976, a bill originating in the House of Representatives was mistakenly submitted to the Governor-General and assented to. [......] The Governor-General revoked the first assent, before assenting to the bill which had actually passed. | revoked the first assent |
| *Perturbed* | What does the Governor-General do for the royal assent? | | NoAns |
| *Original* | What is the Vietnamese word for both blue and green? | In some languages, including old Chinese, Thai, old Japanese, and Vietnamese, the same word can mean either blue or green. [......] Vietnamese uses a single word for both blue and green, xanh. | xanh |
| *Perturbed* | What color are Vietnamese flags, both blue and green? | | NoAns |

Table 9: Additional Examples of *Hard* Unanswerable Questions Generated through Perturbation

## A.2 Randomly-Sampled Examples

| | Question | Context | Self-Labeled Answer |
|---|---|---|---|
| *Original* | How long did the amendment extend the trade agreements? | On 10 January 1941, Germany and the Soviet Union signed an agreement settling several ongoing issues. [......] It also extended trade regulation of the 1940 German–Soviet Commercial Agreement until August 1, 1942. [......] | until August 1, 1942 |
| *Perturbed* | How long will the amendment extend the trade agreements? | | NoAns |
| *Original* | Who are Oklahoma's US Senators? | In the 112th Congress, Oklahoma's U.S. senators were Republicans Jim Inhofe and Tom Coburn, and its U.S. Representatives were John Sullivan (R-OK-1), Dan Boren (D-OK-2), Frank D. Lucas (R-OK-3), Tom Cole (R-OK-4), and James Lankford (R-OK-5). | Jim Inhofe and Tom Coburn |
| *Perturbed* | Who are the US Senators? | | NoAns |
| *Original* | How many 60-gun frigates did the Russians lose in the Black Sea? | During the siege, the Russians lost four 110- or 120-gun, three-decker ships of the line, twelve 84-gun two-deckers and four 60-gun frigates in the Black Sea, plus a large number of smaller vessels. | four |
| *Perturbed* | How many 60-gun warships are in the waters in the Black Sea? | | NoAns |
| *Original* | The second truth is? | The second truth is that the origin of dukkha can be known. Within the context of the four noble truths, the origin of dukkha is commonly explained as craving (Pali: tanha) conditioned by ignorance (Pali: avijja). | the origin of dukkha can be known |
| *Perturbed* | The second question is? | | NoAns |
| *Original* | Buddhism's spread led to what large-scale effort? | In Asia, the spread of Buddhism led to large-scale ongoing translation efforts spanning well over a thousand years. The Tangut Empire was especially efficient in such efforts; exploiting the then newly invented block printing, and with the full support of the government [......] | translation |
| *Perturbed* | What led to what large-scale effort? | | the spread of Buddhism |
| *Original* | What type of theatrical uniforms did Paul VI eradicate from the Vatican? | Paul VI did renounce many traditional symbols of the papacy and the Catholic Church; some of his changes to the papal dress were reversed by Pope Benedict XVI in the early 21st century. Refusing a Vatican army of colourful military uniforms from centuries, he got rid of them. He became the first pope to visit five continents. | army |
| *Perturbed* | What type of theatrical production will Pope Paul visit the Vatican? | | colourful military |
| *Original* | Along with water vapor, what atmospheric substance primarily absorbs the infrared emitted by the Earth? | Earth's surface and the clouds absorb visible and invisible radiation from the sun and reemit much of the energy as infrared back to atmosphere. Certain substances in the atmosphere, chiefly cloud droplets and water vapor, but also carbon dioxide, methane, nitrous oxide, sulfur hexafluoride, and chlorofluorocarbons, absorb this infrared, and re-radiate it in all directions including back to Earth. | cloud droplets |
| *Perturbed* | Along with water, what atmospheric substance also absorbs the heat emitted by the Earth? | | carbon dioxide |
| *Original* | When did pragmatism arise? | In the late 19th and early 20th century several forms of pragmatic philosophy arose. The ideas of pragmatism, in its various forms, developed mainly from discussions between Charles Sanders Peirce and William James when both men were at Harvard in the 1870s. | In the late 19th and early 20th century |
| *Perturbed* | Where did pragmatism arise? | | Harvard in the 1870s |

Table 10: Randomly Sampled Examples Generated through SCENE

## A.3 Types of Synthetic Unanswerable Examples

| Category | Description | Percentage | |
|---|---|---|---|
| | | SCENE | SQuAD 2.0 |
| Negation | Negation word inserted or removed. | 2% | 9% |
| Antonym | Antonym used. | 2% | 20% |
| Entity Swap | Entity, number, or date replaced with other entity, number, or date. | 28% | 21% |
| Mutual Exclusion | Word or phrase is mutually exclusive with something for which an answer is present. | 15% | 15% |
| Impossible Condition | Asks for condition that is not satisfied by anything in the paragraph. | 23% | 4% |
| Other Neutral | Other cases where the paragraph does not imply any answer. | 10% | 24% |
| Answerable | Question is answerable (i.e. dataset noise). | 5% | 7% |
| Ill-formed questions | Question is not well formulated. (i.e. generation noise, only specific to SCENE). | 15% | 0% |

Table 11: Types of negative examples generated by SCENE, compared with SQuAD 2.0 in frequency.

# B Model Details

## B.1 Scientific Artifacts

Regarding softwares, we enumerate their license here. PyTorch (Paszke et al., 2019) was used as the deep learning framework, and is licensed under BSD-3. Transformers library (Wolf et al., 2020) was used for training and evaluation, and is licensed under Apache License Version 2.0. BM25 retrieval was implemented using the Pyserini toolkit (Lin et al., 2021) and is licensed under Apache License 2.0.

Regarding datasets used in this work, we enumerate there license here. SQuAD 1.1 (Rajpurkar et al., 2016) and SQuAD 2.0 (Rajpurkar et al., 2018) are licensed under CC BY-SA 4.0. RTE (Wang et al., 2018; Dagan et al., 2006; Bar-Haim et al., 2006; Giampiccolo et al., 2007; Bentivogli et al., 2009) dataset's license is unknown. BoolQ (Clark et al., 2019) is licensed under CC BY-SA 3.0. BoolQ-3L (Sulem et al., 2022) is licensed under CC BY-SA 3.0. ACE-whQA (Sulem et al., 2021) is licensed under CC BY-SA 3.0.

## B.2 Model Hyperparamters

Models for extractive QA were trained on Nvidia Quadro RTX 6000 GPUs with 24GB GPU Memory, and models for boolean QA and RTE were trained on Nvidia GeForce RTX 2080 Ti GPUs with 11GB GPU Memory. We report hyperparameters for fine-tuning RoBERTa$_{Base}$ across all experiments.

| Hyper-Parameter | Task | | |
|---|---|---|---|
| | SQuAD | BoolQ | RTE |
| train batch size | 32 | 16 | 16 |
| max seq length | 384 | 256 | 128 |
| doc stride | 128 | 128 | / |
| lr_scheduler | Linear | Linear | Linear |
| optimizer | AdamW | AdamW | AdamW |
| adam_beta1 | 0.9 | 0.9 | 0.9 |
| adam_beta2 | 0.999 | 0.98 | 0.999 |
| adam_epsilon | 1e-8 | 1e-8 | 1e-8 |
| num epochs | 3 | 10 | 10 |
| warmup ratio | 0.06 | 0.06 | 0.06 |
| learning_rate | 2e-5 | 1e-5 | 5e-6 |
| weight decay | 0.01 | 0.01 | 0.1 |

Table 12: Hyperparamters for training RoBERTa$_{\text{Base}}$ on various tasks.

**Warm-starting Details.** For warm-starting purposes, $\lambda_{\text{SCENE}}$ is set to 0 in Eq 2 for the first 100 steps, and $\lambda_{\text{SCENE}}$ resumes to 1 for the rest of the training procedure.

### B.3  Computational Cost

| Method | Training Set | Examples (#) | | | Hours | #/sec |
|---|---|---|---|---|---|---|
| | | Real | Synthetic | Total | | |
| No Augmentation | SQuAD 1.1 | 87,509 | 0 | 87,509 | 2 | 12 |
| SCENE (ours) | SQuAD 1.1 | 87,509 | 87,509 $_{\text{Retrieval}}$ + 87,509 $_{\text{Perturbation}}$ | 262,527 | 9 | 8 |
| No Augmentation | SQuAD 2.0 | 130,232 | 0 | 130,232 | 3 | 12 |
| SCENE (ours) | SQuAD 2.0 | 130,232 | 130,232 $_{\text{Perturbation}}$ | 260,464 | 9 | 8 |

Table 13: Expected Computational Cost in Training Our Method (SCENE) Compared with No-Augmentation Baseline on SQuAD. Because SCENE uses the additional BART mask infiller and QQP pre-trained paraphrase detector, it's expected to be slower in training throughput.