# OpenReview forum: "SCENE: Self-Labeled Counterfactuals for Extrapolating to Negative Examples"
_EMNLP/2023/Conference — EMNLP 2023 Main_

### Official Review · Reviewer_oQE5 · 2023-08-02

**Soundness:** 4

**Excitement:**

4: Strong: This paper deepens the understanding of some phenomenon or lowers the barriers to an existing research direction.

**Paper Topic And Main Contributions:**

This paper proposes a novel method for data augmentation (or extrapolation) for negative examples in QA. The authors use self-training and rejection sampling via a paraphrase detector to create negative examples that are useful for training a QA model to accurately detect unanswerable questions without any gold negative examples.

**Reasons To Accept:**

Solid contribution: The proposed method saves on data collection costs and leads to improved performance.

Good writing: The paper is clear and well-written. The authors support their major claims with experimental evidence.

Extensive experiments: The authors provide results across a variety of different QA datasets and see consistent trends showing the effectiveness of their approach.

**Reasons To Reject:**

While not required, it would have been nice to see few-shot performance from an LLM (such as bloom-3b) on these datasets. The authors mentioned in the Limitations section that they specifically wanted to focus on smaller models, but it would be good to know the performance from an LLM since this task seems relatively straightforward for an LLM with a lot of general knowledge.

**Reproducibility:**

4: Could mostly reproduce the results, but there may be some variation because of sample variance or minor variations in their interpretation of the protocol or method.

**Reviewer Confidence:**

3: Pretty sure, but there's a chance I missed something. Although I have a good feel for this area in general, I did not carefully check the paper's details, e.g., the math, experimental design, or novelty.

---

> ### Author Rebuttal · Authors · 2023-08-29
>
> We thank the reviewer for their comments and suggestions. We are glad that the reviewer found our contribution solid, experiments extensive and our paper well-written. We would like to respond to the following comments:
>
> **Comparison with Few-Shot Performance of LLM**: As we mentioned in the Limitations section, compared to LLMs, smaller models are more efficient. Moreover, it can be an unfair comparison with LLMs on open-web datasets like SQuAD due to data contamination concerns when LLMs are trained. Data contamination acknowledgements can be found at Section 8 (on Page 36) from the PaLM paper [1] where SQuAD 2.0 is mentioned specifically as contaminated. We will add related discussions in the final version.
>
> **Reference**: [1] Chowdhery, Aakanksha et al. “PaLM: Scaling Language Modeling with Pathways.” ArXiv abs/2204.02311 (2022). https://arxiv.org/abs/2204.02311

---

### Official Review · Reviewer_NjSu · 2023-08-04

**Typos Grammar Style And Presentation Improvements:** None
**Soundness:** 5

**Excitement:**

4: Strong: This paper deepens the understanding of some phenomenon or lowers the barriers to an existing research direction.

**Missing References:**

None

**Paper Topic And Main Contributions:**

This paper proposes a framework for counterfactual sample generation. The core innovation of this paper is to generate new classes of negative samples compared to the previous methods.

**Questions For The Authors:**

1. It has been emphasized many times in the paper that SCENE is able to construct "harder negative examples", but the description of "harder" is very vague, so can the hardness of the samples be described more accurately? Compared to SCENE, I think the retrieval-based generation method yields syntactically more diverse negative examples, so perhaps SCENE does not produce harder negative examples from the perspective of syntactic diversity.
2. How the values of the hyperparameters $\lambda_{\rm Shuf}$ and $\lambda_{\rm Retr}$ vary in the experiment ? Do they change before and after the warm-starting phase?
3. What is the meaning of the sentence located on lines 392 through 393?

**Reasons To Accept:**

1. This paper is well organized and contains full of content and detailed experiments, obtaining significant improvement under the different settings.
2. SCENE is innovative enough that it removes barriers between datasets with different labels in several types of QA and RTE tasks, making it possible to test the migration of future models between datasets or to integrate multiple datasets into the same form.
3. The problem studied is of high practical value since unanswerable questions are common in the practice of QA, while SCENE can effectively reduce the labeling cost of related tasks

**Reasons To Reject:**

1. There are a lot of detailed parts of the paper, but there are still parts that have not been discussed due to the over-complexity of this pipeline. For example, whether a paraphrase detection model in Filter is necessary and how much removing it affects the results

2. Experimental results show that an increase in performance on the new label brings about a decrease in performance on the original label. And in practice, performance labels on the original label (e.g., the F1 or EM performance of the "Has Answer" part) tend to be more important. So I don't think creating harder samples in the data processing part is the best solution, maybe going over an additional fact-checker (e.g., an LLM) after generating the answer can keep the performance of the model on the original labels drop less.

**Reproducibility:**

3: Could reproduce the results with some difficulty. The settings of parameters are underspecified or subjectively determined; the training/evaluation data are not widely available.

**Reviewer Confidence:**

5: Positive that my evaluation is correct. I read the paper very carefully and I am very familiar with related work.

---

> ### Author Rebuttal · Authors · 2023-08-29
>
> Author Response:
> We thank the reviewer for their comments and suggestions. We are glad that the reviewer found our task valuable, our method SCENE innovative and effective, and our writing well organized. We would like to respond to the following comments and questions:
>
> **Necessity of Paraphrase Detector**: During our ablation study in lines 447-488, we ablated the Paraphrase Detector and called it “No Filter”. In this ablation set up, a generated counterfactual is acceptable if and only if it changes the model prediction (we will elaborate this more in our revision). Removing the paraphrase detector (“No Filter” case) decreases accuracy by 1.9 F1 on SQuAD 2.0 and  5.0 F1 on ACE-whQA. In this regard, we feel having a paraphrase detector in the loop is necessary.
>
> **Decrease of Performance on “HasAns” subset**: This is due the fact that a model trained on SQuAD 1.1 will never predict "unanswerable" whereas any model trained to identify unanswerable questions will sometimes misclassify an answerable question as unanswerable. This happens for all methods of training a model to predict "unanswerable", including training on SQuAD 2.0. Table 2 also supports this: the baseline method “No Augmentation” trained on SQuAD 2.0 has worse performance on answerable examples compared with a model trained on just SQuAD 1.1 and evaluated on SQuAD 2.0. Please also see our explanation for this phenomenon in lines 414-419.
>
> **Using LLM in the loop**:  Admittedly LLMs can fit reasonably well in our filtering process by checking for paraphrases. However, if we try to use LLMs directly to check whether a question is unanswerable, we would need to assume that they are already good at the task – which defeats the purpose of creating a training dataset for detecting unanswerability. Moreover, there can be dataset contamination concerns when LLMs were trained (see Section 8 from the PaLM paper [1] where SQuAD 2.0 is mentioned specifically as contaminated). Nevertheless, we sincerely thank the reviewer for suggesting LLMs and we will consider it for future research.
>
> **[Question 1] Comparison of Counterfactuals Quality between SCENE and Retrieval**: We also compared with a Retrieval-based method, where we match each question to a passage with highest word overlap but not containing the answer for the question within the same dataset, and we label these question-passage pairs as unanswerable questions. In Table 2, we found using our method SCENE is better than Retrieval on SQuAD 2.0.
> We also manually inspected qualities of SCENE-generated counterfactuals and Retrieval-based ones in Appendix A. We found SCENE-generated ones are harder to distinguish answerable from unanswerable than Retrieval-based ones, even though they may not be linguistically “harder” or syntactically diverse. SCENE questions are harder for unanswerable detection tasks because they are minimally edited, and thus they have high lexical overlap with original answerable questions.
>
> **[Question 2 and 3] Hyperparameter Details**: $\lambda_\textrm{Shuf}$ and $\lambda_\textrm{Retrieval}$ are either 1 or 0 based on different experimental settings (see related ablation in Table 11 at Appendix B.1). They stay the same before and after the warm-starting phase.
> Lines 392-393 explain one of our design choices on choosing $\lambda$’s, that we set them either 1 or 0, unless for warm-starting purposes, where some $\lambda$’s are temporarily suppressed to 0. For example, $\lambda_\textrm{SCENE}$ is set to 0 in Eq 2 for the first 100 steps, and it resumes to 1 for the rest of the training procedure.
>
> **Reference**: [1] Chowdhery, Aakanksha et al. “PaLM: Scaling Language Modeling with Pathways.” ArXiv abs/2204.02311 (2022).  https://arxiv.org/abs/2204.02311

---

### Official Review · Reviewer_b1yf · 2023-08-12

**Soundness:** 4

**Excitement:**

4: Strong: This paper deepens the understanding of some phenomenon or lowers the barriers to an existing research direction.

**Paper Topic And Main Contributions:**

This paper introfuces a new method for generating negative samples for QA mthods. This is an important (and often overlookd problem)

**Questions For The Authors:**

3A) How do you measure the quality of the generated counterfactuals? Did you find the need for any sort of filteration?

**Reasons To Accept:**

1) The methods and comparisons made are sound and exhaustive.
2) The results are justified by the explanations.


**Reasons To Reject:**

None, fairly comprehensive work.

**Reproducibility:**

4: Could mostly reproduce the results, but there may be some variation because of sample variance or minor variations in their interpretation of the protocol or method.

**Reviewer Confidence:**

4: Quite sure. I tried to check the important points carefully. It's unlikely, though conceivable, that I missed something that should affect my ratings.

---

> ### Author Rebuttal · Authors · 2023-08-29
>
> We thank the reviewer for their comments and suggestions. We are glad that the reviewer acknowledged the importance of our main goal: generating subtle negative samples for question answering. We are excited to see that the reviewer found our methods sound, comparisons comprehensive and results justified well. We would also like to respond to the following comments:
>
> **Quality of the Generated Counterfactuals**: We measure the quality of generated counterfactuals by how successfully the perturbation changes the model prediction. We also performed manual inspections of our generated counterfactuals in Appendix A, and concluded that our generated counterfactuals are more subtle and harder to detect unanswerability than both Shuffling and Retrieval baselines.
>
> **Necessity of Filtering**: Please see lines 223-254 for the filtering steps, where we use an off-the-shelf paraphrase detector to filter out bad examples. For ablation on different filtering components, please see our ablation section at Lines 447-488 and Table 4, where we found that removing all filtering processes and assuming all generated counterfactuals NoAns decreased the F1 score by 10.5 points on ACE-whQA, compared to our method SCENE. Therefore, we believe the filleting process is necessary.

---

### Meta-Review · Area_Chair_weXF · 2023-09-18

**Recommendation:** 5

**Metareview:**

This paper introduces a novel pipeline to automatically create unanswerable questions given QA examples. The pipeline includes four key components: question perturbation, paraphrase detection, answerability prediction, and filter of unanswerable questions. All reviewers appreciated the well-structured content, innovative methods, and comprehensive experiments.

---

### Decision · Program_Chairs · 2023-10-07

**Decision:**

Accept-Main

**Comment:**

This paper introduces a novel pipeline to automatically create unanswerable questions given QA examples. The pipeline includes four key components: question perturbation, paraphrase detection, answerability prediction, and filter of unanswerable questions. All reviewers appreciated the well-structured content, innovative methods, and comprehensive experiments.